# Risk Factors Associated with Refractory Epilepsy in Patients with Tuberous Sclerosis Complex: A Systematic Review

**DOI:** 10.3390/jcm10235495

**Published:** 2021-11-24

**Authors:** Dominika Miszewska, Monika Sugalska, Sergiusz Jóźwiak

**Affiliations:** Department of Pediatric Neurology, Medical University of Warsaw, 02-091 Warsaw, Poland; dominika.miszewskaa@gmail.com (D.M.); monikaslowinska91@gmail.com (M.S.)

**Keywords:** tuberous sclerosis complex, drug resistant epilepsy, refractory seizures, risk factors

## Abstract

Background: Epilepsy affects 70–90% of patients with tuberous sclerosis complex (TSC). In one-third of them, the seizures become refractory to treatment. Drug-resistant epilepsy (DRE) carries a significant educational, social, cognitive, and economic burden. Therefore, determining risk factors that increase the odds of refractory seizures is needed. We reviewed current data on risk factors associated with DRE in patients with tuberous sclerosis. Methods: The review was performed according to the PRISMA guidelines. Embase, Cochrane Library, MEDLINE, and ClinicalTrial.gov databases were searched. Only full-text journal articles on patients with TSC which defined risk factors related to DRE were included. Results: Twenty articles were identified, with a cohort size between 6 and 1546. Seven studies were prospective. Three factors appear to significantly increase DRE risk: *TSC2* mutation, infantile spasms, and a high number of cortical tubers. Conclusions: A proper MRI and EEG monitoring, along with genetic testing, and close observation of individuals with early onset of seizures, allow identification of the patients at risk of DRE.

## 1. Introduction

Approximately 70–90% of individuals with tuberous sclerosis complex (TSC) experience epilepsy, and in approximately one-third of them, the seizures become refractory to treatment [1,2]. Refractory seizures, also defined as drug-resistant epilepsy (DRE) are commonly defined as “the failure of adequate trials of two tolerated and appropriately chosen and used antepileptic drug (AED) schedules (whether as monotherapies or in combination) to achieve sustained seizure freedom” [3]. It bears a significant educational, social, cognitive, and economic burden.

Tuberous sclerosis complex (TSC) is a genetic disease affecting 1:6000–1:13,000 newborns, caused by a mutation in one of two loci: *TSC1* located on chromosome 9q34.13) or *TSC2* (located on chromosome 16p13.3) [4]. More than 60% of the cases are sporadic de novo mutations, while the remaining 30% are transmitted in an autosomal dominant pattern [5]. *TSC1* and *TSC2* genes mutation inactivate the function of the proteins produced by those genes [6]. The process leads to the mammalian target of Rapamycin (mTOR) overactivation. As mTOR signaling is involved in regulation of many basic cellular processes e.g., proliferation and growth, the mutation causes excessive growth of multiple benign tumors, and alteration in the morphology and function of neurons [7,8].

The current guidelines on TSC diagnosis recommend both clinical and genetic diagnostic criteria, recently updated and published in October 2021 by Northrup et al. [9].

In TSC, the clinical seizures appear mostly between three and five months of age [10]. The first seizures are often subtle and therefore not noticed or confused with physiological behavior. Among patients with TSC, there is a high risk of developing infantile spasms [11]. Those factors negatively impact intellectual development of the individuals, and, according to most caregivers (76.5%) in the TOSCA study, also hinders their family life and social relations [12,13].

Current diagnostic methods such as prenatal echocardiography, brain magnetic resonance imaging (MRI), and EEG monitoring allow an early diagnosis before the onset of clinical seizures enabling early disease-modifying treatment [14]. As a consequence, preventive treatment is becoming more approved and often introduced in clinical centers [15]. Therefore, defining parameters associated with refractoriness, especially those which may be identified early, is necessary. It enables early recognition of patients at-risk, to whom preventive treatment is crucial to facilitate their development.

This article aimed to provide a systematic review of current data regarding risk factors and early predictors of refractory epilepsy in individuals with TSC. It focuses on diagnostic imaging, EEG monitoring, clinical diagnosis, genetic and molecular findings.

## 2. Materials and Methods

To accomplish this systematic review, the Preferred Reporting Items for Systematic Reviews and Meta-Analyses (PRISMA) guidelines [16] were followed, during the design, search, and reporting stages.

### 2.1. Eligibility Criteria

For article selection, we applied the standard PICOS (Population, Intervention, Comparators, Outcome measure, and Study design) criteria. The main hypothesis is that there are risk factors associated with the development of drug-resistant epilepsy (DRE). The definition of DRE differed between studies and is discussed below.

(a)P (patients)—patients with TSC.(b)I (intervention)—development of DRE.(c)C (comparator)—we searched for studies comparing patients with refractory and non-refractory epilepsy.(d)O (outcome)—an association between risk factors and DRE.(e)S (study design)—only full-text, original studies published in English or Polish.

We included only full-text journal articles on patients with TSC assessed based on epilepsy status and response to treatment. Only studies published in English or Polish were included. The search was not limited to any publication date or status.

We excluded articles that did not fulfill the inclusion criteria, studies on animal models or tissues, or non-full-text articles (e.g., conferences’ abstracts).

### 2.2. Information Sources

One of the researchers (DM) searched by electronic databases: MEDLINE, Embase, and Cochrane Library. We also performed a manual search in the references of previously included studies and review articles. Additionally, a search for ongoing or previous trials on the topic was performed on ClinicalTrials.gov. The search was done between 13 May 2021 and 31 May 2021.

### 2.3. Search Strategy

The following search phrases were used to perform the search: (“tuberous sclerosis”) AND ((refractory) OR (“drug resistant”) OR (drug-resistant)) AND ((epilepsy) OR (seizures)). In ClinicalTrials.gov search, done on 31 May 2021, we applied only “tuberous sclerosis” term.

### 2.4. Study Selection

The articles were reviewed upon eligibility assessment according to the three-phase procedure: (1) title, (2) abstract, and (3) full-text analysis. All three phases were performed independently by two authors (DM, MS). The authors evaluated the full text if an abstract met the inclusion criteria but provided insufficient information. The review results were evaluated between the reviewers. Any disagreement was resolved by the discussion and consensus.

### 2.5. Data Collection Process and Data Items

The two reviewers (DM and MS) individually performed screening and selection of data from the articles. The discussion and consensus resolved the differences between the reviewers. To minimize the risk of data duplication, we examined patients’ characteristics and authors’ names. In case of any concern related to the data duplication, the articles were again evaluated and compared by DM.

Data regarding title, author’s name, year of publication, study type, sample size and characteristics, inclusion and exclusion criteria, refractory epilepsy definition, risk factors associated with drug resistance, *p* values for each factor if available were retrieved.

### 2.6. Assessing the Risk of Bias in Individual Studies

To assess the risk of bias, we performed Cochrane risk of bias tool [17] for randomized trials and the Newcastle–Ottawa Scale (NOS) [18] for nonrandomized studies.

NOS scale assigns 0 or 1 point for each answer, in three groups of criteria: (1) Selection: 4 questions, a maximum of 4 stars, including (a) representativeness of the exposed cohort, (b) selection of the non-exposed cohort, (c) ascertainment of exposure, (d) if the outcome of interest was present or not at the start of the study; (2) Comparability of the cohorts on the basis of the study design or analysis: 2 questions, a maximum of 2 stars; (3) Outcome: 3 questions, a maximum of 3 stars, including (a) outcome assessment, (b) whether the follow-up duration was long enough for the outcomes to occur, (c) adequacy of follow-up of cohorts [18]. The studies were grouped based on the score; 9–7, 6–4, and 3–0 stars were defined as low, moderate, and high risk of bias, respectively.

The Cochrane risk of bias assesses the risk of bias in five domains. Domain 1 is the randomization process; domain two is a deviation from the intended interventions; domain 3—missing outcome data; domain 4—measurement of the outcome; and 5—selection of the reported result. In each domain, response options are given to a few signaling questions, and then the risk-of-bias judgment is performed [17]. Based on those responses, the overall risk of bias is estimated as low, high, or some concerns.

The studies were evaluated independently by DM and MS. Disagreements were resolved by the third reviewer or discussion and consensus.

### 2.7. Summary Measures

The primary outcomes in this study were: the association between risk factors and the development of drug-resistant epilepsy. Due to the diversity between risk factors assessed in each study, we reported all the measured parameters. We set a *p*-value of 0.05 or less as statistically significant.

### 2.8. Data Analysis

Data are expressed as a qualitative description of statistically significant risk factors.

## 3. Results

### 3.1. Study Selection and Available Literature

The search of MEDLINE, Embase, Cochrane Library, and ClinicalTrials.gov provided a total of 1187 citations. Additional six citations were supplemented from the references’ lists of included or review articles. 84 duplicates were excluded. In the first and second phases of screening 1059 articles were not included. During full text analysis, additional thirty-one reports were excluded as they either did not meet inclusion or met exclusion criteria (Figure 1). Overall, 19 reports were included in the systematic review [2,10,12,14,19,20,21,22,23,24,25,26,27,28,29,30,31,32,33]. The inclusion and exclusion criteria available for each article are included in Table 1.

### 3.2. Study Characteristics

Studies design: Six of the included studies were prospective, one of which was randomized, and two had randomized control and partially non-randomized open-label groups [24,28]. The remaining 13 studies were based on retrospective data.

Patients: The included studies involved patients with TSC. The number of patients included in each study varied between 6 and 1546 people.

Intervention: Each study compared patients with DRE and those responsive to therapy.

Primary and additional outcome: The included studies reported on the association between DRE and risk factors, and 18 out of the total 19 articles provided *p* value to assess statistical significance.

The definition of drug-resistant epilepsy differed among the studies, and therefore, each is included in Table 1.

### 3.3. Risk of Bias within Studies

Out of sixteen non-randomized studies assessed in the Newcastle–Ottawa Scale, 15 received a total score of either 7, 8, or 9 stars, placing among the low biased group. A study by Monteiro et al. received 5 stars (3 in the selection category, 0 in the comparability, and 2 in the exposure) [27]. Therefore, this article qualified as a moderate risk of bias.

All the randomized studies were assessed as a low overall risk of bias [24,28,33].

The rating of the risk of bias is included in Table 2.

### 3.4. Results of Included Studies

Data extracted from each study are presented in Table 1.

#### 3.4.1. Definition of Drug-Resistant Epilepsy (DRE)

Ten studies (52.6%) used the exact ILAE 2010 definition of drug-resistant epilepsy: “a failure of adequate trials of two tolerated, appropriately chosen and used antiepileptic drug schedules (whether as monotherapies or in combination) to achieve sustained seizure freedom” [3]. Of those, Hulshof et al. specified the threshold age for control as two years [14]. Other studies applied other various definitions of DRE. In four studies, authors did not provide a definition. Detailed criteria for DRE used in particular studies are included in Table 1.

#### 3.4.2. Association of Genetic Mutation and DRE

Ten studies (52.6%) investigated a relationship between genetic mutations and DRE [12,19,20,21,22,23,25,27,28,32]. Six (60%, 6/10) found statistically significant correlation between *TSC2* and DRE. *TSC2* mutation as a risk factor was declared: in two studies when *TSC2* vs. *TSC1* mutations were compared [12,28] and in three studies without specifying the control. Savini et al. performed the study on six patients, and therefore did not estimate the *p* value [25]. Yet, they found a particular mutation variant associated with DRE: in the GAP domain of *TSC2*.

Two studies did not find statistical significance when *TSC1* vs. NMI and *TSC2* vs. NMI were compared [19], one of the articles previously mentioned as the article which observed *TSC2* vs. *TSC1* mutation [12]. The study by Chu-Shore et al., when compared *TSC2* vs. NMI and *TSC2* vs. *TSC1*, did not list it as a risk factor [23]. The last two studies did not find any association between known TSC mutations and DRE.

The impact of family history of TSC on DRE presence was reported in four studies. One study found it to be associated with a lower frequency of refractory seizures [31], and the other that the lack of family history of TSC was linked with DRE [20]. The studies by Mert et al. and Zhang et al., did not encounter statistical significance [26,32].

#### 3.4.3. Type and Time of Seizures and DRE

Three out of four studies (75%) that focused on the history of infantile spasms identified it as statistically significantly associated with DRE. One of them determined drug-resistant infantile spasm as a more potent risk factor. Yet, the duration of infantile spasms does not appear to have an impact [12].

The epilepsy onset age is suggestive of being a parameter related to DRE. One article described the threshold age for the onset of focal seizures before 1 year of age [12], the other one below two years of age [26], and two more did not specify the age [20,21]. The second study also found a stronger association with DRE if the seizures were present in the neonatal period.

In two articles, a history of status epilepticus was associated with DRE risk [21,26].

#### 3.4.4. Psychiatric Disorders and the Risk of DRE

In six studies, more severe cognitive impairment defined as mild to severe intellectual disability was associated with DRE. Lower educational level was observed to be related with refractoriness in two articles [20,21].

Two studies determined autism spectrum disorder as associated with DRE. One of them declared that attention deficit hyperactivity disorder (ADHD) is related to a higher risk of DRE and, on the contrary, anxiety lowers the risk [12]. At the same time, the other one did not find any association with ADHD [26]. Two studies observed a relationship between psychiatric disorders and DRE [20,21].

#### 3.4.5. MRI/CT Changes and DRE

Zhang et al. determined calcification in cerebral parenchyma as a statistically significant (*p* = 0.006) risk factor for DRE [32].

The cortical tubers are known to cause seizures in TSC patients [34]. Six out of eight (75%) studies that analyzed MRI findings determined focal cortical dysplasia (FCD) as associated with DRE. Four of those defined FCD as cortical tubers and ascertained the threshold number of tubers: seven and more cortical tubers [21], four and more [26], “at least one cyst-like cortical tuber” [2]. The last one compared the number of tubers 4.89 vs. 4.41 in DRE and non-refractory epilepsy, respectively, though the difference was not statistically significant [14].

One out of four articles that searched for the association between a presence of subependymal nodules (SEN) or subependymal giant cell astrocytoma (SEGA) and DRE found them to raise the risk of statistical significance.

Two articles analyzed white matter dysplasia and migration lines and did not find any association with DRE [12,26].

#### 3.4.6. EEG Findings and DRE

Mert et al. observed an increased risk of DRE if EEG discharges became generalized [26].

Younger age of first ictal epileptiform discharges (IED) and multifocal IED on the first EEG are associated with refractoriness in patients treated according to the standard protocol, according to de Ridder et al. [33]. The difference was statistically significant only in a univariable model. When put in a multivariable model, or if the patients were preventatively treated, the risk was not statistically significant.

#### 3.4.7. Treatment and the Risk of DRE

Three studies examined whether preventive treatment could be a predictor associated with reduced risk of DRE [24,29,30]. Two of them found a statistically significant difference between the groups [24,30].

## 4. Discussion

The present study aimed to summarize current knowledge on risk factors associated with DRE in patients with TSC. The results were classified into six categories: genetic mutation, time and type of seizures, psychiatric disorders, MRI/CT changes, EEG findings, and treatment protocol. Identifying the parameters related to increased risk of refractoriness, especially those which may be defined early, is crucial. Those factors related to the increased risk of refractoriness have the potential as indicators in finding patients at risk, who are most likely to benefit from early disease-modifying treatment.

Only ten articles had the same standardized DRE definition, based on ILAE, 2010 consensus. Jozwiak et al. in 2019 included in antiepileptic therapies: pharmacotherapy, VNS implantation, ketogenic diet, and epilepsy surgery. Other studies defined the highest acceptable number of seizures in a specific time. Five studies included in this review did not provide any definition of DRE. To minimize the impact of these discrepancies, we decided to include the information on the DRE definition in Table 1.

*TSC2* mutation has been widely described as related to worse clinical outcomes, and the recent EPISTOP study confirmed the previous assumptions [35,36,37,38,39,40,41]. The results of our study also reflect this association, as most of the authors found a correlation between *TSC2* mutation and increased risk of DRE. This pathogenic variant remains a strong risk factor of DRE when compared with *TSC1* and NMI (60%, 6/10). However, some studies pointed out that the gene mutation might have no impact on seizure’s refractoriness [12,19,20,23,32]. It is possible that the risk of DRE may more depend on the particular type of mutation. However, we did not find such detailed analysis of the association between the type of gene mutation and DRE. Although *TSC2* mutation is not always associated with DRE, patients with a mutation in this gene should be considered as having a higher risk of worsening the clinical course of TSC, including the risk of DRE.

A recent study by Liu et al. described a relationship between *TSC1* truncating mutation and intractability. However, it was performed on tissues from TSC patients operated on due to DRE, and we did not include the results in the review [42].

Family history of TSC lowers DRE risk; the difference may be explained by more attentive caregivers trained to early recognize alarming symptoms. Our results are consistent with literature, where family *TSC2* cases tend to be described as less severe than de novo mutations [39]. Importantly, familial cases of TSC are more common to be caused by *TSC1* mutations, which is known to be less severe [39,43].

Regarding epilepsy, most of the studies that investigated infantile spasms found its relationship with DRE. However, there was some strong contra indicatory evidence from Vignoli et al. [21]. West syndrome’s role in developing DRE may be explained by the early onset of seizures and difficulties in diagnosis and introducing proper treatment. Interestingly, the duration of infantile spasms appears not associated with DRE in the study by Jeong et al. [12]. Yet, the authors suggest the results were based on incomplete data, which impaired the correct analysis of this variable.

Younger age of onset of IED and clinical seizures seem to increase the odds of DRE [9,44]. The age threshold remains undefined, with some limiting it to two years, others one year of age or even the neonatal period [12,20,21,26]. Our results are reflected in the other studies’ findings, according to which IED presence in most children is a sign of epileptogenesis and a predictor for refractory epilepsy [33,45,46]. Therefore, frequent EEG monitoring of children with TSC before clinical seizures to introduce preventive treatment is currently recommended [9,24,25,47,48]. Many centers already implement the early EEG and preventive treatment in TSC patients, based on the results of clinical trials performed in the last decade and European recommendations [48,49]. According to Słowińska et al., half of the treatment centers (31/60, 51.7%), introduce treatment based on EEG findings prior to clinical seizure onset [48,49].

Some studies included in this review also showed that early or even preventive antiseizure treatment of patients with TSC may reduce the risk of DRE [24,30,50]. Therefore, early TSC diagnosis and proper education for TSC patients’ custodians become crucial in early and effective treatment. According to the EPISTOP study, preventive treatment is related to a significant reduction of DRE risk compared to the introduction of treatment after clinical seizures (28% vs. 64%, respectively) [24]. Currently, European recommendations advocate the introduction of preventive antiseizure treatment in children within 24 months of life if ictal discharges occur on EEG, with or without clinical manifestation [47]. On the other hand, recently updated international recommendations also notice potential benefits of preventive treatment; however, the consensus committee determined that additional evidence is needed before preventative treatment with vigabatrin can be recommended for all infants with TSC universally [9]. A recent questionnaire study showed that preventive approach is becoming more and more widely implemented in clinical practice [48].

Interestingly, de Ridder et al. found that once preventive treatment is implemented, none of the factors which had increased the risk of DRE (younger age of the first IED on EEG and multifocal IED on the first EEG) remained significant [33]. EEG abnormalities have been recently considered as a biomarker of epileptogenesis in infants with TSC [45,46]. In our review, Mert et al. observed an increased risk of DRE in case of generalized discharges on EEG [26]. EEG is a non-invasive and, in many centers, easily available study. Therefore, it may be used as a valuable for the early detection of patients with increased risk of DRE. Regular EEG studies within first 24 months of life in patients with TSC are currently recommended by international and European recommendations [9,47].

Primary evidence from the recent EXIST-3 trial on everolimus confirms that early introduction of this mTOR inhibitor decreases DRE risk [51,52]. In addition, some former smaller studies demonstrated its positive effect on cardiac rhabdomyomas, SEGA size, and epilepsy in TSC patients, including as an adjunctive treatment for DRE [51,53,54,55].

Intellectual disability is linked with DRE, yet it appears to be the result, not the cause of DRE [56,57,58]. Early-onset of severe epilepsy is known to hinder the intellectual development of the patients and lead to lowered IQ [44,59,60,61]. The results of our review also reflect this association, as Goh et al. found a relationship between intellectual disability and infantile spasms, while according to Winterkorn et al., the cognitive outcome is related to DRE and *TSC2* mutation [31,62]. Age of onset, *TSC2* mutation, and infantile spasms are independently related to DRE, as described in our article. Therefore, we may assume that refractory seizures impair the cognitive and intellectual development of TSC patients.

The relationship between other psychiatric disorders, including ASD and ADHD, and DRE remains unclear. Specchio et al. suggested both ASD and DRE be driven by FCD [63]. Other studies found that the early onset of seizures and infantile spasms to be the causes of autism development [60,64,65]. It appears that the same factors cause ASD and DRE or that ASD takes origin in DRE. Interestingly, anxiety and depression continue without any association with refractoriness.

MRI or CT imaging findings appear to be related to DRE. Sixty-six percent of the studies which investigated FCD found it to increase the risk of DRE. Though the difference is not statistically significant in one of them, we included the results due to its specific limitations, such as fetal MRI quality and the study’s retrospective design [14]. A study on resected cortical tubers and perituberal cortex by Ruppe et al. has demonstrated the epileptogenic potential of both [66]. The abnormalities in the proximity of ventricles, such as SEN and SEGA, and white matter disruption seem unrelated to epileptogenesis.

The relationship between sleep—its quality, duration, any disturbances, and seizures, is well-known in many epileptic syndromes [67,68,69]. Few studies indicated the increased risk of sleep disturbances in patients with TSC [70,71]. However, none of the analyzed articles discussed the influence of sleep on DRE in individuals with TSC. As sleep disturbances present a potentially modifiable factor it seems to be reasonable to include them onto the list of possible risk factors of DRE in TSC in future studies.

The general characteristics of the patients, such as sex, age, and race, show no association with refractoriness in the included studies [12,20,26,32]. The male to female ratio appears to be maintained at an equal level. Age is a risk factor only in specific circumstances, e.g., the onset of seizures or the onset of IED on EEG. Both instances were discussed above.

### Limitations of the Study

Limitations of the search and selection: The search of the articles—was conducted only by one reviewer. The risk of bias may be higher than if the search was performed by two reviewers separately.

We included only articles published in English or Polish. Therefore, some articles written in other languages may have been omitted.

Due to differences in DRE definitions, neither comparison between the studies nor a metaanalysis were performed.

The limitation at the outcome level: Most of the included studies were retrospective. In one study, only six patients were included. Therefore, the risk of bias is higher compared to randomized and prospective studies. Limitations of the presented systemic review: The main limitation of this systematic review is a low number of high-quality data from randomized studies. Moreover, studies differed, i.e., in terms of the applied definition of DRE. Therefore, due to the risk of bias, no metanalysis or comparison between the studies was conducted.

## 5. Conclusions

Most studies observed an association between DRE and three main parameters: *TSC2* mutation, infantile spasms, and the number of cortical tubers. According to the authors, epileptiform discharges on EEG and early onset of seizures, especially before one year of age, also increase the risk of refractoriness of the seizures. The majority of the risk factors is unmodifiable, yet regular EEG monitoring and proper education of the caregivers was observed to reduce the risk of refractoriness.

Psychiatric disorders, such as ADHD, ASD, and cognitive impairment appear to be the consequence rather than the cause of DRE.

Importantly, all three studies, which focused on preventive treatment, observed lowered DRE risk if the treatment was introduced before clinical seizures.

This study summarizes current knowledge on risk factors related to increased risk of DRE in individuals with TSC. The results facilitate identifying patients with the highest odds of developing refractory seizures. In those individuals, an introduction of treatment before clinical seizures may contribute to their developmental improvement.

## Figures and Tables

**Figure 1 jcm-10-05495-f001:**
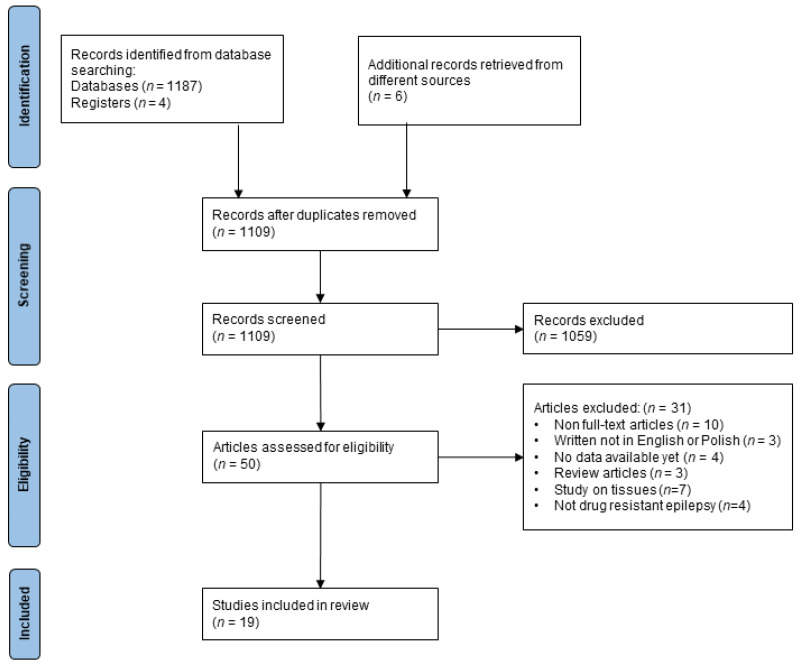
Flow diagram of the study selection.

**Table 1 jcm-10-05495-t001:** Characteristics of the cohorts in the included studies.

Author, Year	DRE ^1^ Definition	Participants Included	Inclusion and Exclusion Criteria	Factors Associated with DRE	Factors Not Associated with DRE
Benova et al., 2018 [22]	Authors did not provide DRE definition. However, the following variables were considered markers of DRE: number of AED ^2^ used number of AED at the end of follow-up the absence of seizure-free status at the end of follow up	22	Inclusion: pre/perinatal diagnosis of cardiac rhabdomyomas	Higher number of areas with FCD-like ^3^ features (uncorrected *p* < 0.0001, FDR ^4^ < 0.01) ID ^5^ (uncorrected *p* < 0.001, FDR < 0.05) *TSC2* (uncorrected *p* < 0.01, FDR < 0.05)	-
Chu-Shore et al., 2009 [2]	uncontrolled seizures after more than three AED (not including treatment for infantile spasms)	173 (2 months to 73 years, median 13 years)	-	At least one cyst-like cortical tuber (*p* = 0.0007) FCD	-
Chu-Shore et al., 2010 [23]	uncontrolled seizures after at least three first-line AED trials	291	-	Infantile spasms (*p* < 0.0001)	*TSC2* vs. *TSC1*, *TSC2* vs. NMI ^6^ (*p* = 0.169)
Hulshof et al., 2021 [14]	ILAE, 2010 ^7^, at 2 years	41	Inclusion: Fetal MRI of sufficient quality and available neurologic outcome data at the age of 2 years Exclusion: Epilepsy surgery before the age of 2 years.	-	Fetal (sub)cortical lesion sum score—4.89 vs. 4.41 in DRE and non-refractory epilepsy, respectively (*p* = 0.62)
Jeong et al., 2017 [12]	ILAE, 2010	1546 (9.6 to 25.5 years, median 16.0 years) 21.4%—*TSC1* 67.9%—*TSC2* 10.7%—NMI	Exclusion: if date fields were missing and age of onset and symptom duration could not be calculated	Onset of focal seizures prior to 1 year of age (*p* < 0.001) *TSC2* (*TSC2* vs. *TSC1* (*p* < 0.001)) Infantile spasms (*p* < 0.001) Drug-resistant infantile spasms (*p* < 0.001) ASD ^8^ (*p* < 0.001) Mild to moderate intellectual disability - ID (*p* < 0.001) and severe to profound ID (*p* < 0.001) ADHD ^9^ (*p* < 0.001) Anxiety (*p* = 0.02) Periungual fibromas (*p* = 0.02)—lower odds of DRE	Male vs. female (*p* = 0.94) Race (*p* = 0.40) *TSC2* vs. NMI (*p* = 0.84) *TSC1* vs. NMI (*p* = 0.12) Duration of infantile spasms (*p* = 0.90) Depression (*p* = 0.08) SEGA ^10^, SEN ^11^, cortical tubers, cerebral white matter migration lines Anxiety after adjusting for TSC mutation (*p* = 0.69)
Jozwiak et al., 2011 [30]	two or more seizures per month despite the use of two or more AED	45—total 35—standard treatment (AEDs within a week after the onset of seizures), 14—preventive treatment (AEDs within a week after appearance of active epileptic discharges on consecutive EEG, but before clinical seizures)	Inclusion: Diagnosis of TSC until the end of second month of life, follow-up till the end of 24 month of life Exclusion: children presenting with seizures	Standard treatment vs. preventive treatment (*p* = 0.021)	-
Jóźwiak et al., 2019 [29]	two or more seizures a month despite the use of two or more antiepileptic therapies, including AEDs, ketogenic diet, vagus nerve stimulation, and epilepsy surgery	39—total 25—standard treatment (vigabatrin within a week after first clinical seizures), 14-preventive treatment (vigabatrin introduced within a week after epileptiform discharges, before clinical seizure).	Inclusion: Diagnosis of TSC until the end of second month of life, follow-up till the end of 24 month of life Exclusion: children presenting with seizures	-	Standard treatment vs. preventive treatment (*p* = 0.5)
Kotulska et al., 2014 [10]	ILAE, 2010	21	Inclusion: Epilepsy onset within 4 weeks of life.	Presence of FCD	-
Kotulska et al., 2021 [24]	ILAE, 2010	94 (both groups underwent careful EEG surveillance)	Inclusion: TSC diagnosis within first 4 months of life, no history of clinical seizures or epileptiform abnormalities in EEG.	Lower odds of DRE if preventive treatment (*p* = 0.047)	-
Mert et al., 2019 [26]	seizures once a month or more for at least 1 year, while using at least two AED at the appropriate dose	83	Inclusion: At least 1 year follow-up.	Seizures in the neonatal period Age of onset of seizure less than 2 years of age ASD Status epilepticus Infantile spasms Generalization of EEG finding Tuber count of more than 3 (*p* < 0.001) IQ < 70	Sex Consanguinity Family history of TSC Attention-deficit and hyperactivity disorder SEN SEGA White matter dysplasia (*p* > 0.05)
Monteiro et al., 2014 [27]	ILAE, 2010	35	*-*	*TSC2* mutation	-
Ogórek et al., 2020 [28]	ILAE, 2010	94	Inclusion: Age ≤ 4 months, no prior seizures, no clinical seizures on baseline video EEG Exclusion: any condition considered by the investigator to hinder participation in the study or affect primary outcome.	*TSC2* (*TSC2* vs. *TSC1* mutation (*p* = 0.0245))	-
Peron et al., 2018 [19]	-	240	Inclusion: 0–80 years of age, conventional molecular analysis available for both *TSC1* and *TSC2*, complete clinical and imaging data available and updated to the latest follow-up encounter. Exclusion: (1) Possible clinical diagnosis or (2) Insufficient clinical records.	-	*TSC1* vs. NMI (*p* = 1) *TSC2* vs. NMI (*p* = 0.7)
Savini et al., 2020 [25]	-	6	-	ID Pathogenic variants in the GAP domain of *TSC2* (no *p*-value, just case reports)	-
de Ridder et al., 2021 [33]	ILAE, 2010	83—total 51—standard (S; clinical and EEG follow-up and start of vigabatrin after seizure onset) 23—preventive (P; follow-up and introduction of vigabatrin once EEG criteria met—focal IED for >10% of the recording time, multifocal IED, generalized IED, or hypsarrhythmia—and before seizure onset)	-	S group: Younger age of first IED ^12^ on EEG (*p* = 0.019). Multifocal IED on the first EEG compared to focal IED (OR 4.4, 95% CI 1.1–16, *p* =0.026).	S group: Younger age of first IED on EEG in a multivariable model (*p* = 0.429). Multifocal IED on the first EEG compared to focal IED in a multivariable model (*p* = 0.058). P group: None of the features of the first EEG with epileptiform discharges.
Vignoli et al., 2013 [21]	ILAE, 2010	160	Inclusion: At least 1 year follow-up	Cognitive impairment (*p* < 0.05) *TSC2* mutation More than 6 cortical tubers SEN or SEGA Lower educational level Psychiatric disorder Earlier mean age of epilepsy onset (3.3 vs. 5.3 years, *p* > 0.05) Status epilepticus (*p* < 0.05) Younger age at TSC diagnosis (7.6 vs. 13.2 years, *p* < 0.05)	Infantile spasms (*p* > 0.05) Epilepsy onset in the first year of life
Vignoli et al., 2021 [20]	ILAE, 2010	257 (>18 years old)	-	ID (*p* < 0.001) Psychiatric disorders (*p* = 0.004) No family history of TSC (*p* = 0.010) Younger age of seizure (6 vs. 27 months, *p* = 0.001) Higher rate of spasms (27.1% vs. 48.8%, *p* = 0.007) Less frequently focal epilepsy (*p* = 0.029) Lower level of education (*p* = 0.002)	Age Sex Mutation Tubers SEN
Winterkorn et al., 2007 [31]	one of the following criteria met: more than three AED, epilepsy surgery was performed, or one or more seizures per day continued despite therapy	208	-	Family history of TSC—lower odds of DRE (*p* = 0.003) low IQ/DQ (*p* < 0.0005)	-
Zhang et al., 2018 [32]	ILAE, 2010	108 (3 months to 10 years, mean 2.2 years, median 1.4 years)	Inclusion: Taking rapamycin > 1 year	Calcification in the cerebral parenchyma (*p* < 0.006)	Patient’s age (*p* = 0.745) Seizure type (*p* = 0.788) Genetic mutation (*p* = 0.204) Family history (*p* = 0.927)

^1^ DRE—Drug-resistant epilepsy, ^2^ AED—antiepileptic drugs, ^3^ FCD—Focal cortical dysplasia, ^4^ FDR—False Discovery Rate correction from univariate tests, ^5^ ID—Intellectual disability, ^6^ NMI—No mutation identified, ^7^ “Drug-resistant epilepsy is defined as failure of adequate trials of two tolerated, appropriately chosen and used antiepileptic drug schedules (whether as monotherapies or in combination) to achieve sustained seizure freedom”. In [3] ^8^ ASD—Autism spectrum disoder, ^9^ ADHD—Attention deficit hyperactivity disorder, ^10^ SEGA—Subependymal giant cell astrocytomas, ^11^ SEN—Subependymal nodules, ^12^ IED—ictal epileptiform discharges.

**Table 2 jcm-10-05495-t002:** Methodic assessment of the included studies.

Author, Year	Study Design	Risk of Bias Assessment
	The Newcastle–Ottawa Scale	The Cochrane Tool
	Selection (0–3)	Comparability (0–2)	Outcome (0–3)	Total (Risk of Bias)	Risk of Bias
Benova et al., 2018 [22]	prospective	4	2	2	8 (Low)	
Chu-Shore et al., 2009 [2]	retrospective, comparative	4	2	3	9 (Low)	
Chu-Shore et al., 2010 [23]	retrospective comparative	4	2	3	9 (Low)	
Hulshof et al., 2021 [14]	retrospective cohort	4	2	3	9 (Low)	
Jeong et al., 2017 [12]	retrospective, multicenter, from TSC Natural History Database Project	4	2	3	9 (Low)	
Jozwiak et al., 2011 [30]	prospective, nonrandomized clinical trial	4	2	3	9 (Low)	
Jóźwiak et al., 2019 [29]	prospective, nonrandomized clinical trial	3	2	3	8 (Low)	
Kotulska et al., 2014 [10]	retrospective	4	2	3	9 (Low)	
Kotulska et al., 2021 [24]	multicenter, prospective, randomized clinical trial and partially open-label	-	-	-	-	Low
Mert et al., 2019 [26]	retrospective	4	2	3	9 (Low)	
Monteiro et al., 2014 [27]	retrospective	3	0	2	5 (Moderate)	
Ogórek et al., 2020 [28]	randomised control and non-randomised open-label	-	-	-	-	Low
Peron et al., 2018 [19]	retrospective	4	2	3	9 (Low)	
Savini et al., 2020 [25]	retrospective	3	1	3	7 (Low)	
de Ridder et al., 2021 [33]	multicenter, prospective, randomized	-	-	-	-	Low
Vignoli et al., 2013 [21]	retrospective	4	2	3	9 (Low)	
Vignoli et al., 2021 [20]	retrospective	4	2	3	9 (Low)	
Winterkorn et al., 2007 [31]	retrospective	4	2	3	9 (Low)	
Zhang et al., 2018 [32]	retrospective	4	2	3	9 (Low)

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
