# Peer review of "Risk Factors Associated with Refractory Epilepsy in Patients with Tuberous Sclerosis Complex: A Systematic Review"

_jcm, 2021, doi:10.3390/jcm10235495_

Round 1

Reviewer 1 Report

Manuscript: 1444223

 Risk Factors Associated with Refractory Epilepsy in Patients 2 with Tuberous Sclerosis Complex: A Systematic Review

The present study aimed to summarize current knowledge on risk factors associated  with Drug-resistant epilepsy (DRE)in patients with TSC.

Most studies observed an association between  DRE and three main  parameters: TSC2 mutation, infantile spasms, and the number of cortical tubers. According to the authors, epileptiform discharges in EEG and early onset of seizures, especially before one year of age, also increase the risk of refractoriness of the seizures. The majority of the  risk factors is unmodifiable, yet regular EEG monitoring and proper education of the caregivers was observed to reduce the risk of refractoriness.

The article is well done and interesting. In relation to the importance of modifiable factors it would be useful to evaluate the aspect of sleep, given the important relationship between epileptic discharges and sleep disturbances. Have you found any data on this? Discuss this point.

Author Response

Response to Reviewer 1 Comments

Point 1: The present study aimed to summarize current knowledge on risk factors associated  with Drug-resistant epilepsy (DRE)in patients with TSC.

Most studies observed an association between  DRE and three main  parameters: TSC2 mutation, infantile spasms, and the number of cortical tubers. According to the authors, epileptiform discharges in EEG and early onset of seizures, especially before one year of age, also increase the risk of refractoriness of the seizures. The majority of the  risk factors is unmodifiable, yet regular EEG monitoring and proper education of the caregivers was observed to reduce the risk of refractoriness.

The article is well done and interesting.

Response 1: Thank you for your time and favorable review.

Point 2: In relation to the importance of modifiable factors it would be useful to evaluate the aspect of sleep, given the important relationship between epileptic discharges and sleep disturbances. Have you found any data on this? Discuss this point.

Response 2: Thank you for this interesting comment. Certainly, the relationship between sleep (such as its quality and duration) and seizures is well-known in many epileptic syndromes. However, none of the analyzed studies discussed the influence of sleep on epilepsy, including DRE, in patients with TSC; therefore, we did not encompass this parameter in our review.

There are few studies indicating the increased risk of sleep disturbances in patients with TSC. (Zambrelli, E., et al.. (2021). Sleep and behavior in children and adolescents with tuberous sclerosis complex. American journal of medical genetics. Part A, 185(5), 1421–1429.) and (van Eeghen, A. M, et al. (2011). Characterizing sleep disorders of adults with tuberous sclerosis complex: a questionnaire-based study and review. Epilepsy & behavior : E&B, 20(1), 68–74.).

We agree with the reviewer that this potential association should be studied in future studies on TSC. Therefore, we included this comment in the discussion in lines 370-375. Thank you.

Reviewer 2 Report

Manuscript: Risk Factors Associated with Refractory Epilepsy in Patients with Tuberous Sclerosis Complex: A Systematic Review (Manuscript Number jcm-1444223)

This is a comprehensive, systematic review of the English language and Polish language literature regarding risk factors for drug-resistant epilepsy (DRE) in patients with tuberous sclerosis complex (TSC).  It condenses into clinically meaningful aliquots the available literature on factors predicting or correlated with DRE in TSC in the areas of genetic mutations, time and type of seizures, psychiatric disorders, neuroimaging findings, EEG findings, and treatment protocol. 

The Materials and Methods as well as the Results are thorough and well-organized.

In the Discussion, the paragraph on lines 255-7 is redundant following the previous paragraph and should be removed.

In the Conclusion lines 380-381, it is stated that “an introduction of treatment before clinical seizures would contribute to their developmental improvement.”  This needs to be clarified because it is unclear whether this statement refers to the EXIST-3 trial on everolimus or to the two out of three studies that found preventative antiseizure medication treatment to reduce risk of DRE.  If this is referring to the latter, it would be better to change this to read “an introduction of treatment before clinical seizures may contribute to their developmental improvement.”

There are small grammatical errors that will need to be corrected (e.g., “free-dom” on line 179, no space after the end of the sentence on line 306, “EEG abnormalities has” rather than “have” on line 316). 

Author Response

Response to Reviewer 2 Comments

Point 1: This is a comprehensive, systematic review of the English language and Polish language literature regarding risk factors for drug-resistant epilepsy (DRE) in patients with tuberous sclerosis complex (TSC).  It condenses into clinically meaningful aliquots the available literature on factors predicting or correlated with DRE in TSC in the areas of genetic mutations, time and type of seizures, psychiatric disorders, neuroimaging findings, EEG findings, and treatment protocol. The Materials and Methods as well as the Results are thorough and well-organized.

Response 1: Thank you for your time and favorable review.

Point 2: In the Discussion, the paragraph on lines 255-7 is redundant following the previous paragraph and should be removed.

Response 2: Thank you for this comment. We addressed the issue and removed the paragraph.

Point 3: In the Conclusion lines 380-381, it is stated that “an introduction of treatment before clinical seizures would contribute to their developmental improvement.”  This needs to be clarified because it is unclear whether this statement refers to the EXIST-3 trial on everolimus or to the two out of three studies that found preventative antiseizure medication treatment to reduce risk of DRE.  If this is referring to the latter, it would be better to change this to read “an introduction of treatment before clinical seizures may contribute to their developmental improvement.”

Response 3: Thank you for this valuable point. We agree with the reviewer and implemented the suggested changes in the manuscript.

Point 4: There are small grammatical errors that will need to be corrected (e.g., “free-dom” on line 179, no space after the end of the sentence on line 306, “EEG abnormalities has” rather than “have” on line 316).

Response 4: Thank you for pointing this out. According to your remark, we performed an additional language check and implemented the required modifications.